# Fully Endoscopic Spine Separation Surgery in Metastatic Disease—Case Series, Technical Notes, and Preliminary Findings

**DOI:** 10.3390/medicina59050993

**Published:** 2023-05-21

**Authors:** Kajetan Latka, Waldemar Kolodziej, Kornel Pawlak, Tomasz Sobolewski, Rafal Rajski, Jacek Chowaniec, Tomasz Olbrycht, Masato Tanaka, Dariusz Latka

**Affiliations:** 1Department of Neurosurgery, St. Hedwig’s Regional Specialist Hospital, ul.Wodociagowa 4, 45-221 Opole, Poland; 2Department of Neurosurgery, Institute of Medical Sciences, University of Opole, Al.Witosa 26, 45-401 Opole, Poland; 3Department of Radiotherapy, Opole Center of Oncology, ul.Katowicka 66a, 45-061 Opole, Poland; 4Department of Orthopaedic Surgery, Okayama Rosai Hospital, Okayama 702-8055, Japan

**Keywords:** spine endoscopy, spine separation, spine metastases

## Abstract

*Objective*: This report aims to describe the surgical methodology and potential effectiveness of endoscopic separation surgery (ESS) in patients with metastatic spine disease. This concept may reduce the invasiveness of the procedure, which can potentially speed up the wound healing process and, thus, the possibility of faster application of radiotherapy. *Materials and Methods*: In this study, separation surgery for preparing patients for stereotactic body radiotherapy (SBRT) was performed with fully endoscopic spine surgery (FESS) followed by percutaneous screw fixation (PSF). *Results*: Three patients with metastatic spine disease in the thoracic spine were treated with fully endoscopic spine separation surgery. The first case resulted in the progression of paresis symptoms that resulted in disqualification from further oncological treatment. The remaining two patients achieved satisfactory clinical and radiological effects and were referred for additional radiotherapy. *Conclusions*: With advancements in medical technology, such as endoscopic visualization, and new tools for coagulation, we can treat more and more spine diseases. Until now, spine metastasis was not an indication for the use of endoscopy. This method is very technically challenging and risky, especially at such an early stage of application, due to variations in the patient’s condition, morphological diversity, and the nature of metastatic lesions in the spine. Further trials are needed to determine whether this new approach to treating patients with spine metastases is a promising breakthrough or a dead end.

## 1. Introduction

Spinal metastases are a common and debilitating complication in patients with advanced cancer, with an estimated frequency of 30–70% among cancer patients [1]. Approximately 10% of them present neurological symptoms [2]. The most commonly affected region is the thoracic spine, followed by lumbar and cervical [3]. The prognosis of patients with spinal metastases can vary widely, depending on factors such as primary cancer type, tumor biology, presence of visceral metastases, and the patient’s overall performance status [3].

Generally, the survival of patients with spinal metastases can range from several months to several years [4]. In the past, due to limited diagnostic capabilities, these patients often presented to surgeons with severe neurological conditions, leaving therapeutic options largely restricted to simple palliative decompression surgeries. In some cases, extensive corpectomy was performed, usually with a high risk of complications [4].

Thanks to improved access to imaging diagnostics, especially magnetic resonance imaging (MRI), metastatic lesions can now be detected at earlier stages of progression. There has also been rapid development in radiotherapy, particularly stereotactic body radiotherapy [5]. The high therapeutic efficacy of these methods has significantly reduced the need for extensive resection procedures. The primary benefit of stereotactic body radiotherapy (SBRT) is for patients with oligometastatic disease. In this patient group, SBRT can improve overall survival for up to 2 years [6]. Additionally, in the palliative setting for symptom control in patients with painful spinal metastases, stereotactic body radiotherapy is superior to conventional radiotherapy in improving the complete response rate for pain [7].

In some cases, where the tumor compresses or is in direct contact with nerve structures, separation surgery may be necessary. This procedure involves removing enough tumor volume to create a several-millimeter gap between the tumor and nervous structures, allowing for safe radiation. Spinal endoscopy has rapidly developed in recent years. With the introduction of new tools and highly effective coagulation, the list of indications for this technique is constantly expanding [8]. Therefore, in this study, we utilized a novel method to visualize the separation procedure in patients with metastatic lesions in the spine. The rationale for using such a minimally invasive technique is that the wound healing time is shorter, which should accelerate the initiation of radiotherapy following surgery.

## 2. Materials and Methods

The described series of cases was developed based on our team’s assumption that using the endoscopic technique to perform separation procedures would be beneficial in strictly selected cases.

Inclusion criteria: 1/ patients with known primary disease or multiple spinal lesions suggesting a metastatic process; and 2/ systemic contraindications to long-term anesthesia and the risk of hemodynamic disturbances associated with extensive open surgical procedures.

Exclusion criteria: 1/ suspected primary neoplastic process; and 2/ rapidly progressing debilitating symptoms requiring extensive decompression.

The procedure has been performed in three patients so far. The Spinal Instability Neoplastic Score (SINS) [9] was assessed and exceeded 7 points in each case. The radiological status was evaluated using the Epidural Spinal Cord Compression (ESCC) scale [10].

Patients 2 and 3 were medically burdened and advanced in age, making open surgery too risky for them. The first patient, despite being younger, was initially in a very severe neurological condition.

We assessed the patients’ neurological status before and after the procedure, blood loss during the procedure, postoperative radiographs, hospital stay duration, and the time between surgery and radiotherapy initiation.

Operation procedures: Each procedure was performed by the same surgeon experienced in performing endoscopic procedures in patients with degenerative diseases. All cases were performed on the thoracic spine with the patient in the prone position under general anesthesia.

The posterior approach was used with an endoscope featuring an eight-degree viewing angle and an 8 mm working canal, standardly used in decompression procedures (Joimax, Karlsruhe, Germany). In all procedures, we utilized a spinal endoscopic irrigation pump and a bipolar coagulation system. The endoscopic irrigation pump allowed for continuous irrigation of the surgical field, ensuring clear visualization and reducing the risk of bleeding. Simultaneously, the bipolar coagulation system, consisting of a typical bipolar electrode used in endoscopy (Joimax, Karlsruhe, Germany), connected to a generator (Bovie, Clearwater, FL, USA), helped achieve effective hemostasis by coagulating bleeding vessels while minimizing the risk of collateral tissue damage.

We have not yet encountered a hemorrhage that was unmanageable by coagulation or one that would make it impossible to continue the procedure. Neuromonitoring was not used during the procedure.

One of the most challenging aspects of this procedure is the initial phase, namely the introduction and docking of the access system. Due to the potential spread of the tumor to the posterior column of the spine, it is crucial to avoid aggressive dilator insertion, which could lead to inadvertent penetration into the spinal canal and subsequent neurological damage. Therefore, meticulous preoperative planning is essential to identify a suitable fulcrum, preferably consisting of healthy bone. Computed tomography and intraoperative X-ray imaging can aid in determining the optimal starting target, typically the area of the intervertebral joint. Once the endoscope is inserted and initial hemostasis is achieved, the planned bone component should be visualized. Tumor-altered tissues are then carefully removed, with ongoing coagulation of any minor bleeding. Bilateral resection is extended until a satisfactory visualization of free nerve structures is attained after exposing the dural sac.

Additionally, tissue differentiation within the tumor is particularly challenging due to the disrupted anatomy at the operative site. In the first phase of the operation, tumor debulking is performed by palpating and removing soft tissue elements using endoscopic instruments. Once a clear view of the dural sac is obtained, the surgeon carefully navigates along its length, expanding the range of decompression both upwards and downwards using a high-speed diamond drill, including the resection performed beneath the dural sac. This is continued until a visual confirmation of healthy bone is achieved. The extent of tumor involvement can be assessed using intraoperative X-ray imaging, capturing images with the endoscope positioned at the extreme ends of the surgical field, i.e., the upper and lower poles. Subsequently, the endoscope is removed, and in the second stage, a short one- or two-sided percutaneous transpedicular fixation is performed using the conventional technique.

Case presentation:

Case no. 1:

A patient in his 50s, without any prior medical history, was diagnosed in the department of neurology due to sudden, deep lower paraparesis. A spine MRI confirmed the presence of a proliferative process localized within almost the entire thoracic spine with significant stenosis at the level of Th6/7 (Figure 1). It was classified as grade 3 compression on the ESCC scale, with a SINS score of 9 points. In the neurologic examination, the Medical Research Council scale for muscle strength (MRC) in the lower extremities was 1/2 for almost a week. Endoscopic separation was performed at the Th6/7 level. After the procedure, an image of a free dural sac was obtained, and its continuity was not broken. However, the patient’s clinical condition did not improve, and the paresis deepened to MRC 1. This was most likely the result of a momentary compression of the spine during the docking of the endoscopic system. The follow-up MRI showed a slight improvement in spinal cord compression, but it was not a fully satisfactory result (Figure 2). It was classified as grade 2 on the ESCC scale. In the opinion of the radiotherapist, the scope of resection allowed the use of SBRT, but due to the severe neurological condition and multiple lesions in the entire spine, the patient was disqualified from further treatment, especially as the histopathological result confirmed advanced multiple myeloma—a diagnosis that only allows for medical treatment. The duration of the surgery was 130 min.

Case no. 2:

An 82-year-old patient was diagnosed with prostate cancer. For 2 weeks, he complained of severe pain syndrome of the thoracic spine with paraparesis MRC 4. A CT scan showed a metastatic lesion involving the right pedicle of the Th2 vertebra, modeling the dural sac at this level (Figure 3). It was 10 points on the SINS score. The patient was qualified for separation surgery with unilateral percutaneous fixation. The procedure was performed using endoscopic visualization. The material was successfully collected, and a free, pulsating dural sac was visualized at the end. There were no bleeding problems during the procedure that would restrict visualization. After the procedure, the patient reported a marked reduction in pain. He was transferred for further oncological treatment. The results of the histopathological examinations confirmed the prostate cancer metastasis. Due to the patient’s age and general condition, further treatment was palliative in accordance with his wishes. We do not have control MRI images. The duration of the surgery was 120 min.

Case no. 3:

A 75-year-old female patient with a history of colorectal cancer 6 years earlier was diagnosed in the neurology department due to severe pain in the thoracic spine that made it impossible for her to stand up or move. The patient had normal muscle strength in her lower limbs, and an MRI of her spine revealed a single, massive hyperplastic lesion located in the right pedicle and vertebral body of Th6 (Figure 4). The lesion was penetrating the spinal canal and causing modeling of the meningeal sac, resulting in a SINS score of 8 points and a grade 2 classification in the ESCC scale. Additionally, a chest X-ray showed a tumor in the right lung.

After consultation with a radiotherapist, the patient was qualified for the separation procedure, and an endoscope was inserted from the rear by anchoring it to the right Th6 pedicle (Figure 5 and Figure 6). After initial coagulation and field cleaning, the boundaries of the Th6 lamina were visualized (Appendix A). Material was collected from the disintegrating, brittle lesion, coagulating on a current basis, and was removed until the dural sac appeared. The decompression was then extended from the level of the upper edge of the Th6 pedicle to the border of the lower boundary plate using a high-speed drill (Appendix A). The entire width of the pulsating dural sac was clearly visible (Appendix A). After the separation was completed, percutaneous Th4/5/7/8 stabilization was performed on the left side only (Figure 7). In the authors’ opinion, unilateral stabilization was preferred due to the reduced amount of artifacts caused by the presence of titanium. Given the patient’s survival time, the biomechanics of such a construct for an oncologic patient were considered secondary. The duration of the surgery was 90 min.

The control CT examination showed a satisfactory degree of bone decompression (Figure 8), and the MRI examination (Figure 9) revealed an image of a several-millimeter gap between the meningeal sac and the hyperplastic lesion, which enabled SRS (Figure 10). This was classified as 1b on the ESCC scale. Blood loss during surgery was minimal, and the patient was able to stand 3 h after the procedure and reported only moderate back pain with a VAS2. She was discharged two days after the procedure and underwent MRI and CT examinations for radiotherapy planning purposes one week later. Registration between planning CT and pre-op and post-op MRI was made and used to determine the target and organs at risk, and an SBRT plan was calculated. The patient was irradiated with 6MV FFF photons in three 8 Gy fractions and was treated for five days with good early tolerance. The histopathological and immunochemistry examinations confirmed a metastatic lesion from colorectal cancer. After spine SBRT lung tumor diagnostic procedures (CT and biopsy), further oncological treatment is planned.

## 3. Results

The intraoperative view of a free pulsating dural sac was achieved in each case. In the first and third cases, where control MRI was available, we were able to achieve an improvement in radiological images. However, the operation on the first patient did not yield the expected clinical results, and further oncological treatment was not possible. In the other two cases, we were able to reduce pain and improve the neurological condition, enabling further oncological treatment. We did not encounter any problems with visualization of anatomical structures due to excessive bleeding, contrary to expected difficulties. Blood loss was minimal in all patients and did not exceed 100 mL. None of the patients required hospitalization for more than the standard 4-day period. Using endoscopic access in cancer patients is challenging, and therefore, cases must be selected individually and be well-planned.

## 4. Discussion

For several years, extensive surgical procedures have been largely abandoned in favor of minimally invasive techniques, and this trend has been observed in virtually all areas of surgery, including spine surgery. With the development of various techniques and novel operating equipment, Kambin et al. (1973) and Hijikata et al. (1975) decided to use an endoscope for discectomy surgery, and after many years, endoscopic resection of spine tumors was attempted [11].

The spine is one of the most common sites for metastases, particularly in its thoracic segment [12]. Anatomically, metastases can be divided into intramedullary, extra-spinal-intradural, and epidural [13]. Possible therapeutic interventions for spine tumors include open surgery, minimally invasive spine surgery, systemic therapy, and radiotherapy [14].

Surgical methods remain an extremely important treatment option, particularly for patients with incurable pain, spine instability, or failed chemotherapy and radiotherapy [15]. Thanks to the availability of numerous surgical techniques such as open surgery, video-assisted thoracoscopic surgery (VATS), percutaneous transpedicular screw fixation (PPSF), minimally invasive spine surgery (MISS) [16], and full-endoscopic surgery of the spine (FESS) [17], it is possible to choose the best method for each patient individually.

The selection of the surgical technique is associated with many factors, such as the patient’s general condition, the experience of the center and the operator in a given surgical technique, anticipated postoperative complications, and the impact on the overall treatment process [18].

For many years, open methods were the only means of treating neoplastic lesions of the spine, which offered significant advantages, such as quick spinal cord decompression, particularly thanks to posterior-access laminectomy. Radical en bloc resection allowed for the removal of significant tumor fragments, and in the case of primary tumors, it allowed for complete recovery. However, open surgery often resulted in significant defects in spine stability, high blood loss, postoperative complications, and a large postoperative wound [16,19,20]. Additionally, it delayed the use of radiotherapy, and studies have shown that abandoning open surgery in favor of stereotactic radiotherapy seems to have greater benefits for the patient. Nonetheless, other studies report significant benefits for patients undergoing radiotherapy who previously underwent open surgery.

Despite the controversy, the combination of open or minimally invasive surgery and radiotherapy is now considered the gold standard in treating spine cancers [21,22]. Percutaneous transpedicular screw fixation (PPSF) is not strictly a resection method because its main goal is to stabilize the spine before and because of pathological fractures. Intraoperative X-ray imaging enables good stabilization results while reducing postoperative complications. It also allows for control of tumor progression by immobilizing the adjacent vertebral bodies [23,24].

Minimal invasive spine surgery (MISS) perfectly fits the current trend of minimizing surgical accesses and represents the natural evolution of open procedures. Dedicated surgical microscopes significantly reduce skin incision while increasing the precision of resection and limiting damage to small vascular and nerve structures [25,26]. An important advantage is significantly lower incidence of complications, less blood loss, lower risk of intensive care, and shorter overall hospital stay [27]. Moreover, thanks to the use of MISS, it was possible to implement radiotherapy very quickly. In the study by Massicotte et al. [28], patients who underwent MISS could undergo stereotactic body radiation therapy (SBRT) after just one week.

Total endoscopic spine surgery (FESS), the theme of our article, is poorly described in the scientific literature. Its main application was in intervertebral disc herniation. Significant advantages of this technique include a low infection rate, lower postoperative opioid usage, and shorter hospital stay [29,30]. Blood loss is lower compared to open techniques [31]. However, there are reports that it is higher than in MISS [32]. It also has greater overall survival than open techniques [17,33].

Separation procedures of spinal metastases have been adequately described in the literature [33]. However, there are very few reports on the use of an endoscope for this purpose. The first report in the literature in this area was the publication by Rosenthal et al. from 1996 [34]. According to their development, the endoscope can be effectively used in spinal oncology for vertebrectomy, vertebra reconstruction, and stabilization. Over the years, further surgical accesses for the endoscope were developed, such as posterior accesses [35,36]; attempts were made to perform subsequent endoscopic procedures, such as corpectomies [37]; and open surgeries were supported endoscopically [38].

The results of endoscopic treatment show great promise, although there are some reports that question its effectiveness [17]. The constantly growing number of new scientific reports creates a very promising future for endoscopic spine surgery [39,40]. However, the low prevalence of this technique and the lack of training centers make it difficult to learn and train. It is estimated to be most widespread in Asia, particularly in South Korea [41]. The increasing interest in minimally invasive surgical techniques creates great opportunities for endoscopic techniques, and the steadily growing number of publications continues to support their efficacy and benefits for the patient.

While the benefits of minimally invasive spine surgery (MISS) and endoscopy are well-documented, it is crucial to consider the potential drawbacks, risks, and complications associated with these procedures, especially in oncological cases where tumors may adhere to the dural sac, complicating decompression. Full-endoscopic spine surgery has a steeper learning curve and requires specialized training to minimize the risk of complications [17]. Limited visualization can be addressed by using high-definition cameras, proper illumination, and adequate irrigation [17].

To avoid excessive spinal cord manipulation in tight tumoral adherences, surgeons should employ meticulous dissection techniques and avoid aggressive retraction of the dura, which can reduce the risk of dural tears. Achieving complete decompression may be more challenging in cases with extensive epidural tumor involvement; however, careful planning based on preoperative imaging can help determine the surgical approach and the extent of decompression [18]. Hemostasis can be more challenging in full-endoscopic spine surgery, particularly in hypervascular tumors. Employing various techniques, such as bipolar coagulation devices and radiofrequency ablation, can help manage hemostasis effectively. By considering these factors and implementing appropriate measures, surgeons can optimize patient outcomes and reduce complications associated with full-endoscopic spine surgery in oncological cases.

The use of endoscopy in oncological procedures remains a controversial topic in the spine surgery community. While the use of an endoscope in separation procedures as preparation for SRS has not been reported in the literature, a good report exists on the use of transforaminal access as palliative decompression in patients with radicular conflict caused by neoplastic proliferation [40].

According to the latest guidelines of the Polish Society of Spine Surgery, in the case of metastatic lesions, the least invasive solution should be used to decompress, stabilize the spine, and prepare the patient for further stereoradiotherapy [42]. This idea guides us towards the use of the endoscopic technique, which we consider to be a less invasive method that not only allows for the collection of more material for histopathological examination than a standard needle biopsy but also decompresses nerve structures. Thanks to the minimal blood loss and potential use of regional or local anesthesia [43], this technique provides an opportunity to treat older and more burdened patients for whom general anesthesia would be too risky.

Effective hemostasis is essential in full-endoscopic spine surgery, particularly when dealing with hypervascular epidural metastases to avoid residual epidural hematoma in cervical and thoracic regions [44]. Several technical recommendations can be employed, such as utilizing a bipolar coagulation device to coagulate bleeding vessels while minimizing collateral tissue damage [45], employing radiofrequency ablation to reduce bleeding [46], administering local vasoconstrictive agents like epinephrine, and using controlled hypotensive anesthesia to decrease blood loss [47]. Additionally, careful sequential dissection and coagulation, adequate irrigation with saline solution, preoperative embolization for highly vascular tumors, optimal patient positioning to reduce venous congestion, and utilizing a hemostatic matrix [48] can all contribute to effective hemostasis. By incorporating these techniques, surgeons can better manage hemostasis during full-endoscopic spine surgery, reducing the risk of residual epidural hematoma and improving patient outcomes. It remains debatable whether this solution is sufficient to perform wide enough decompression. Intraoperative assessment of the degree of decompression is also controversial. Our assumption was that the degree of satisfactory decompression should be like that achieved in degenerative endoscopy—free pulsating nerve structures —and this was the image we achieved in each case. However, radiological images show that the burden was not as extensive as we expected during the procedure. As seen from our first and third cases, the scope of decompression does not seem to be too extensive; however, the presence of blood and postoperative lesions slightly distorts the objective assessment of the control MRI.

The selection of less vascular tumors for full-endoscopic spine surgery (FESS) is crucial to the success of the procedure. In our study, this was not a qualifying criterion; however, it should have been. It is essential to plan and carefully analyze radiological tests to indirectly determine the level of vascularity of a tumor. In all three cases, we were fortunate that bleeding did not pose a significant obstacle during the procedure. Nonetheless, future studies should incorporate this criterion to ensure a safer and more effective FESS procedure. By selecting less vascular tumors, surgeons can minimize the risk of bleeding and improve patient outcomes. Therefore, the accurate assessment of tumor vascularity through radiological analysis should be a priority when considering patients for FESS.

Another controversy is the potential dissemination of tumor cells during lesion resection caused by the flushing pump. Currently, there are no literature reports on this subject. In our opinion, this technique can be used as an alternative to further palliative therapy in burdened patients in advanced stages of the disease.

When selecting patients for FESS, it is crucial to consider factors such as age, tumor burden, frailty index, expected survival time, Karnofsky score, ASA score, and Surgical Risk Scale. In our study, we limited ourselves to more stringent patient selection criteria, but it is potentially possible to expand the selection process to include patients with better health and longer expected survival times. Nevertheless, it is essential to adequately assess the benefits and risks associated with FESS compared to open surgery, taking into account more extensive decompression and tumor resection in these patients. Therefore, further research and larger patient samples are needed to determine whether FESS is a suitable option for patients with better health and longer survival times or if they should continue to undergo open surgery.

This study presents a retrospective review of cases successfully treated by the authors. However, several limitations should be taken into consideration when interpreting the results. Firstly, the retrospective design is subject to potential biases such as selection bias and may not be as reliable as a prospective study. The limited sample size may also hinder the ability to generalize the findings to a larger population or draw definitive conclusions. Moreover, the single-center experience and the absence of a control group make it difficult to directly compare the outcomes of the full-endoscopic technique with other surgical approaches. The use of subjective outcome measures, such as pain and functional scores, may introduce variability and potential bias in the assessment of the results. Finally, the short follow-up period may be insufficient for evaluating the long-term outcomes, complications, and recurrence rates associated with full-endoscopic spine surgery for oncological indications. Future studies should aim to address these limitations through more rigorous research designs, larger sample sizes, multi-center collaborations, and the inclusion of control groups to provide more comprehensive evidence on the safety and efficacy of full-endoscopic spine surgery in oncological cases.

## 5. Conclusions

Rapid advancements in medical technology, such as endoscopic visualization and new coagulation tools, have expanded the treatment options for various spine diseases. Metastatic disease management is multidisciplinary, necessitating adjuvant therapies for effective control. Separation procedures have become the new standard in preparing patients for further radiotherapy and systemic treatment.

Previously, spine metastasis was not considered an indication for endoscopy. Due to the diverse morphology and nature of metastatic lesions in the spine, this technique is technically challenging and risky, particularly in its early stages of application. Endoscopic access for spine metastases should be recommended on an individual basis and be thoroughly planned. For oncologic patients, the focus should be on clinical outcomes rather than wound size or muscle and blood sparing, as seen in degenerative disease patients.

Further trials are needed to determine the true potential of this new method in treating patients with spine metastases. While our results show that accessing these cases is possible, the expected benefits require investigation in a larger and more diverse sample population.

## Figures and Tables

**Figure 1 medicina-59-00993-f001:**
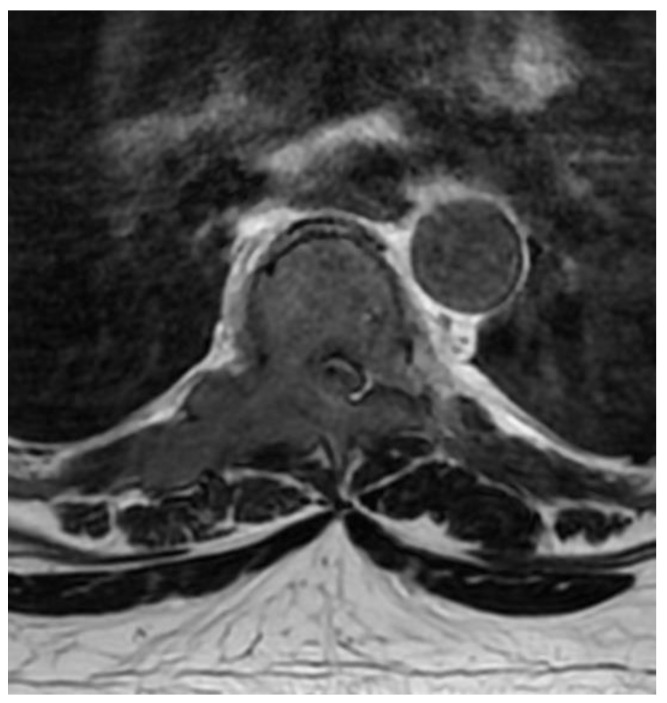
Preoperative T2 axial MRI Th6.

**Figure 2 medicina-59-00993-f002:**
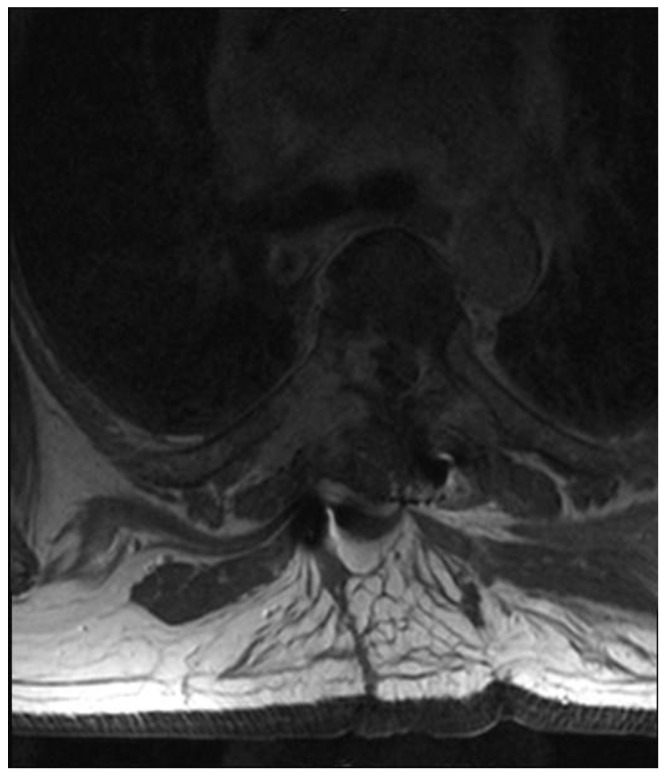
Postoperative T2 axial MRI TH6.

**Figure 3 medicina-59-00993-f003:**
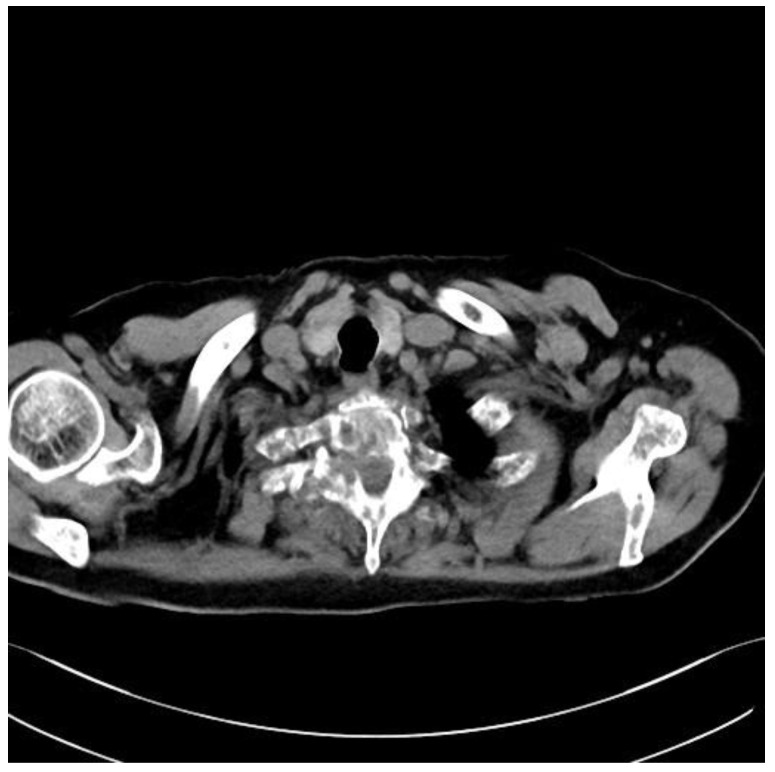
Preoperative axial CT Th2.

**Figure 4 medicina-59-00993-f004:**
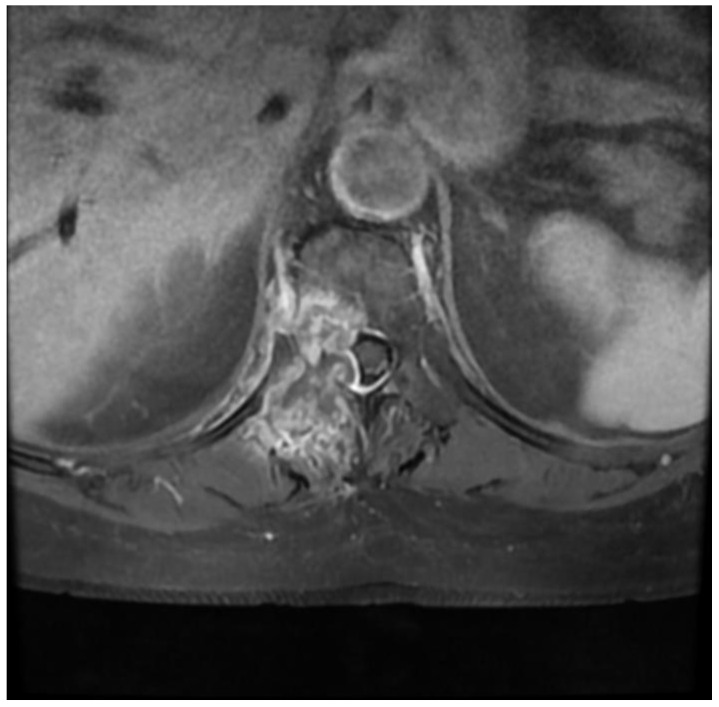
Preoperative T1 contrast-enhanced axial MRI Th6.

**Figure 5 medicina-59-00993-f005:**
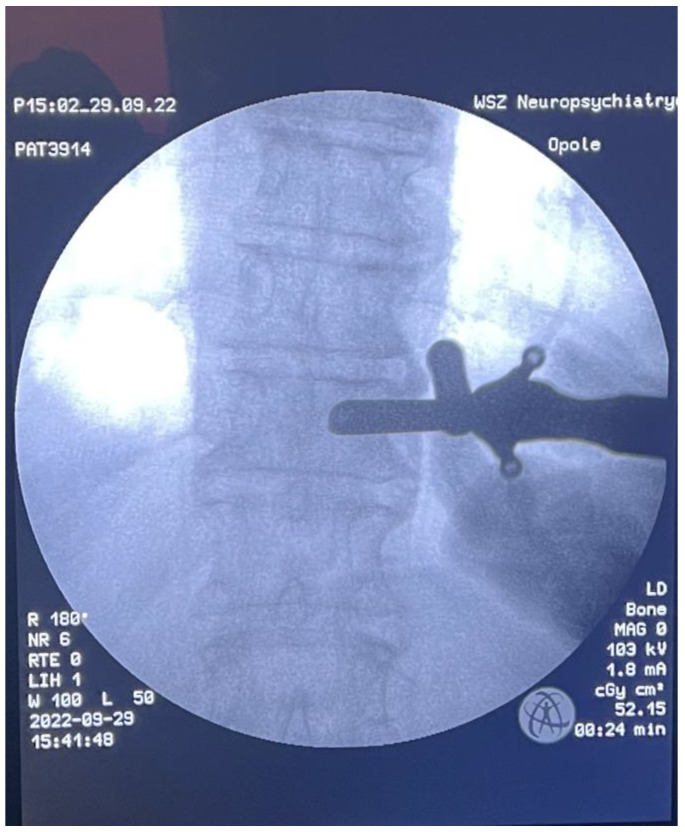
Intraoperative AP X-ray-tube docking point.

**Figure 6 medicina-59-00993-f006:**
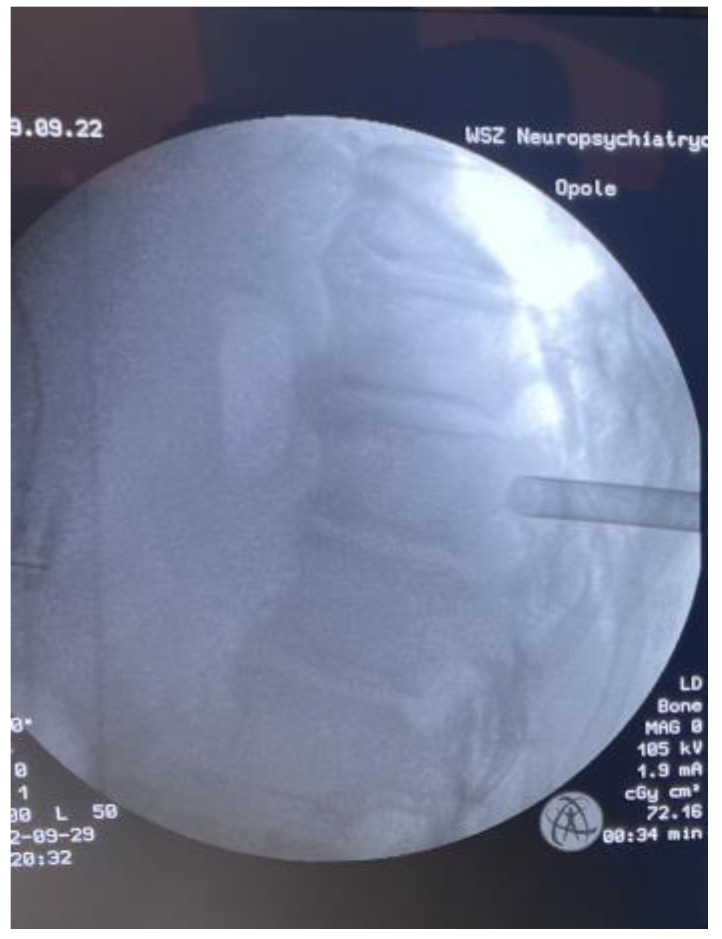
Intraoperative LAT X-ray.

**Figure 7 medicina-59-00993-f007:**
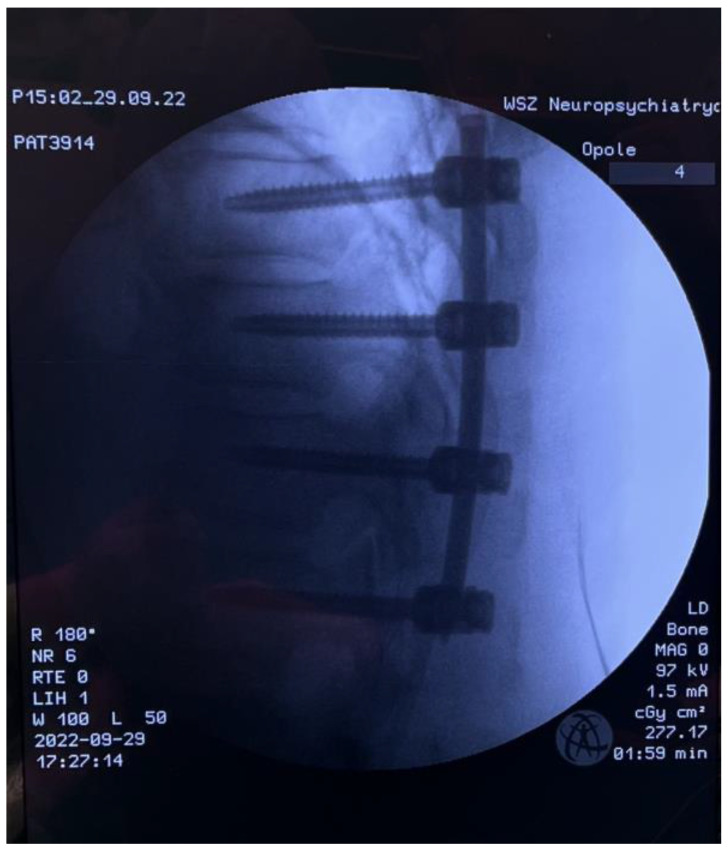
Intraoperative LAT X-ray; percutaneous pedicle screw fixation.

**Figure 8 medicina-59-00993-f008:**
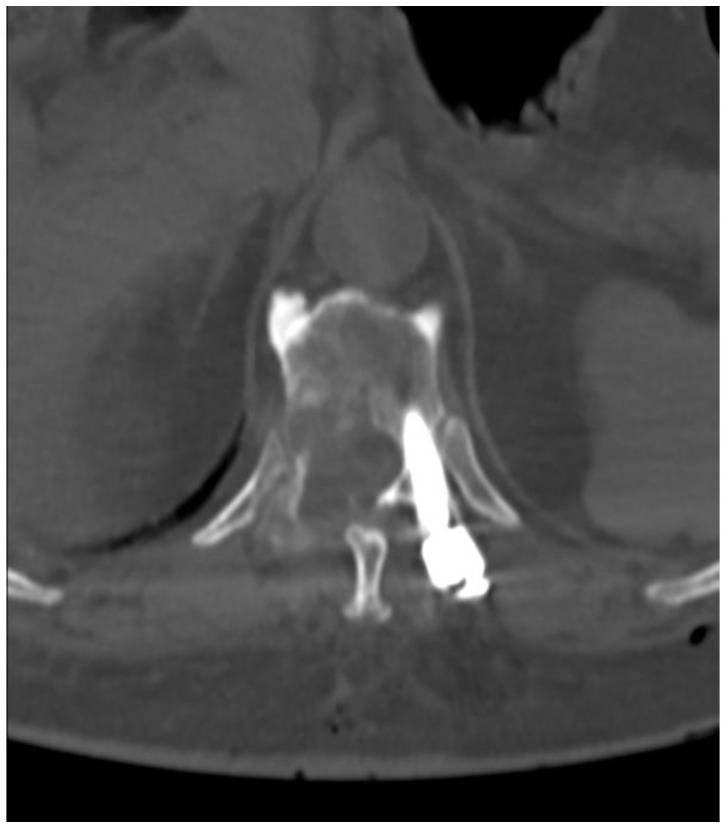
Postoperative axial CT scan of TH6.

**Figure 9 medicina-59-00993-f009:**
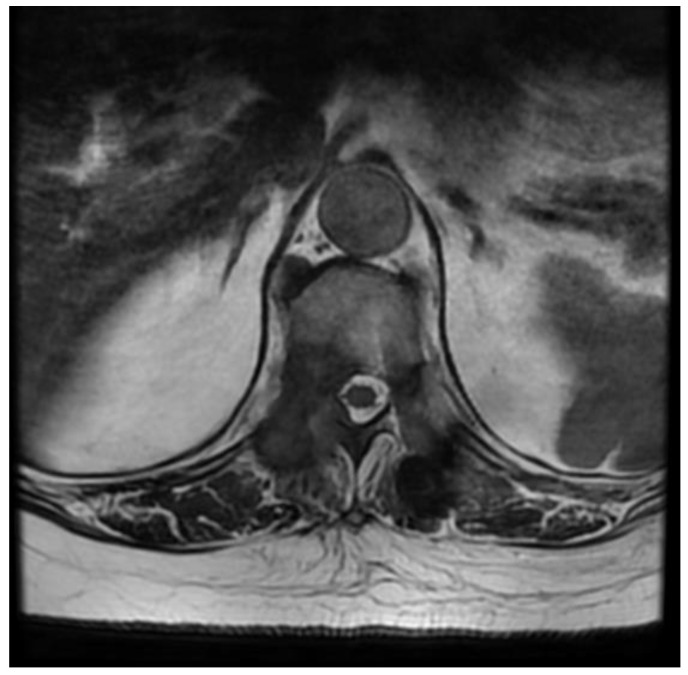
Postoperative axial T2 MRI scan of TH6.

**Figure 10 medicina-59-00993-f010:**
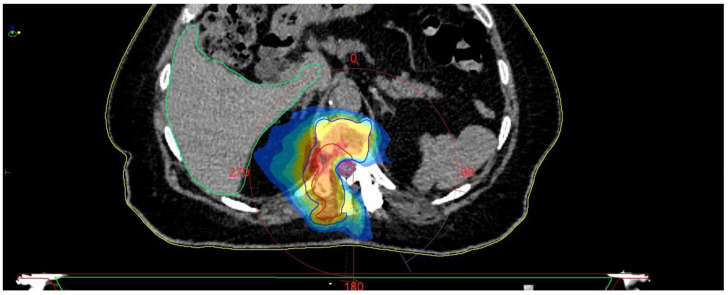
SBRT planning CT.

## Data Availability

Not applicable.

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
