# Peer review of "Fully Endoscopic Spine Separation Surgery in Metastatic Disease—Case Series, Technical Notes, and Preliminary Findings"

_medicina, 2023, doi:10.3390/medicina59050993_

Round 1
Reviewer 1 Report (Previous Reviewer 1)
Although the supplemental video is good, I have several comments that could improve its presentation quality for a scientific journal.
1.- An audio description can replace the background music.
2.- It is essential that the video be concrete and resumed, demonstrating necessary steps of no more than 3 minutes.
3.- The authors can add labels to highlight landmarks or to allow the orientation or identification during the procedure; cranial, caudal, midline, lateral, dural sac, etc.
4.- The font must be discreet, indicating seriousness and methodological rigor.
5.- What is the red arrow trying to indicate?
6.- Surgeons with experience in endoscopy can understand the steps shown in the video. However, those without experience may find it difficult to understand.
7.- The image of the approach, where the authors have placed the working cannula, must be at the beginning, before or after the fluoroscopic confirmation of the working cannula docking.
8.- Specifically, the video must also have an order: clinical presentation, preop images, possible alternatives to endoscopy, approach (fluoroscopic confirmation and intraoperative image), procedure (describing it and pointing out the relevant anatomy), final intraoperative images, postoperative images, and conclusion.
9.- What is the reason for adding a new co-author?
10.- In the discussion, the authors refer to the term Asia-Korea; it should be more precise, for example, use South Korea to differentiate between North Korea, especially for readers without experience in the subject.
11.- The authors should add information on hemostasis during the procedure, such as hypervascular epidural metastases and how to avoid a residual epidural hematoma, of particular interest in the cervical and thoracic region. What technical recommendations exist in full-endoscopy to solve this situation?
12.- Most of the discussion is about the benefits of MISS and endoscopy; however, drawbacks, risks, complications, and how the surgeon could try to avoid excessive spinal cord manipulation in tight tumoral adherences to the dural sac or the endpoint of the decompression with endoscopy? You have pointed out that full endoscopy for oncology differs entirely from degenerative indication.
12.- This manuscript is the retrospective review of a series of cases successfully treated by the authors. Still, it has limitations inherent to its methodology, technique, surgeon, etc... Therefore, the authors should add ¨¨Limitations section¨ during or after the discussion.
10.- The rest of the manuscript has been improved, and I recommend its publication after addressing the recommendations made.
Author Response
Thank you for your valuable feedback!
1-8: The video file has been re-edited (except for the background music).
-
We are pleased to share that Professor Tanaka, our mentor with extensive experience in publishing, has joined the team for final editing and review.
-
The error has been corrected.
-
We have incorporated the aspects of hemostasis into the discussion section.
-
The discussion section has been rewritten accordingly.
-
We have added a section on limitations to the discussion.
- Whole manuscript has been reedited and checked by native
Reviewer 2 Report (New Reviewer)
Introduction; It could also mention the prognosis and frequency of spine metastasis.
Material and methods; Idea should be a clear, better description of the procedure; for example, mention the coagulation system you use. Could you improve the description of the surgery?
The biomechanical justification why did you utilize screws on only one side.
The conclusion Is too long, and you are not able to distinguish it from the discussion.
Author Response
Thank you for your valuable feedback!
- The introduction has been rewritten to provide a better understanding of the study.
- We have made an effort to describe the procedure in more detail and reedited the video clip to improve clarity.
- In our team's opinion, in oncological cases, single-sided fixation with less titanium is preferred to reduce artifacts in control imaging and make SBRT planning easier. While biomechanics is important in some cases, it is not considered a top priority in oncological cases due to the expected survival time. We have added this information to the Materials and Methods section.
- The conclusion has been rewritten and shortened for better readability.
- Whole manuscript has been reedited and checked by native
Reviewer 3 Report (New Reviewer)
This FESS is mainly for debulking and creating a separation between tumor-spinal cord interface for safer post decompression XRT.
1. The selection of patients for FESS is not clearly defined. Age, tumor burden, frailty index, expected survival time, Karnofsky score, ASA score, Surgical Risk Scale can play a factor in patient selection. Can the selection process extended to more healthy patients with longer survival expectancy ? Or should these patients still need open surgery for more extensive decompression and tumor resection ?
2. It was unclear how the author surgeon able to discern between tumor and spinal cord under endoscopic resection.
3. It was obvious that selecting less vascular tumor for FESS was paramount to the success. How was the surgeon able to predict and select less vascular tumors for this trial ?
4. How can the surgeon determine intraoperatively that adequate decompression and separation have been achieved ?
5. What was the duration of these procedure ? Prolong anesthesia posed a increase risk burden as much as in open surgery.
Author Response
Thank you for your feedback.
We have taken your comments into consideration and made the following changes:
- We have included the aspect of selecting less vascular tumors for FESS in the discussion section.
- We agree that it is difficult to predict the degree of vascularity of the tumor, especially at the beginning of the procedure. However, we have emphasized the importance of careful planning and radiological analysis to indirectly assess the degree of vascularity.
- We acknowledge that the selection of less vascular tumors should have been a qualifying criterion in our study, and we have included this in the discussion.
- We have addressed the controversial issue of intraoperative assessment of the degree of decompression, and emphasized our assumption of achieving similar decompression to that in degenerative endoscopy.
- We understand your concern regarding the duration of the procedures(we've added it to case description). We agree that the times reported in our study were relatively long, but we would like to emphasize that these were our first cases, and we were still in the learning process.
- Whole manuscript has been reedited and checked by native
Thank you again for your valuable feedback.
Round 2
Reviewer 1 Report (Previous Reviewer 1)
The video continues with the background music, but it has been summarized, and some references have been added that allow the reader to orient himself. But there are some mistakes, for example, "Th6 metastatic leasion." Nevertheless, the rest of the manuscript has improved, and I consider it can be published after minor video revision.
Author Response
Thank you very much for your thorough review of our manuscript and for providing us with valuable feedback. We greatly appreciate your recognition that the manuscript has improved and can be published after minor revisions to the video. We will carefully address the mistakes and revise the video accordingly. Once again, thank you for your time and effort in reviewing our work.
Reviewer 3 Report (New Reviewer)
The authors described their adaption of endoscopic spine procedure to palliative debulking and decompression of spinal metastasis in selected patients. However, the series is not large enough to convince a wide adoption of this technique for spinal tumor decompression.
Author Response
Thank you for your feedback. We appreciate your comments regarding the size of the series presented in our article. We understand that our study is not aimed at demonstrating that endoscopic procedures in oncological patients will be a new standard in treatment. Instead, our goal was to present a series of three cases showing that such a procedure is possible in selected cases and can have varying outcomes, as demonstrated by the results of our three patients. This article is merely a signal and a guide for future endoscopic surgeons who will attempt to address this challenging topic. We hope that our experience will contribute to further research and larger patient samples to provide more comprehensive evidence on the safety and efficacy of full-endoscopic spine surgery in oncological cases.
This manuscript is a resubmission of an earlier submission. The following is a list of the peer review reports and author responses from that submission.
Round 1
Reviewer 1 Report
Even the authors have made a fantastic effort to show us their surgical technique for separation surgery using endoscopic procedures. I would like to observe the images before my resolution because this uploaded file contains no images.
Author Response
Thank you so much for your kind words. Please see the attached file.
I would like to point out that the scope of resection in the case of the first patient leaves much to be desired and, in my opinion, is a warning against careful qualification for this method, which I tried to emphasize in the publication.

Reviewer 2 Report
1. I can not find the figures in the manuscript.
2. The inclusion and exclusion criteria were not mentioned. This is important to do careful patient selection in the controversial issue.
3. I did not find innovative concept or robust evidence to support the application of Endoscopy in spinal metastasis treatment in the study with limited case report. I suggest the authors revise their paper to include more data and then resubmit it.
Author Response
- the figures were added at the end of the draft. There is a movie from the 3rd case that is included in additional files.
- we've added the inclusion criteria in materials and methods
- I completely agree that the earlier version of the discussion did not include a rational explanation as to why endoscopy makes sense at all in oncology cases. We expanded the discussion in such a way as to justify our reasoning in the choice of therapy, and again drew attention to the appropriate selection of patients for the method. Please check the attachment with the modified draft

Round 2
Reviewer 2 Report
Thanks to the authors for revising their previous manuscript and providing figures. However, for the following reasons, I recommend that the authors resubmit their paper after extensive revision.
The first and most important thing is the study design. The authors submit their paper as the original article. However, this paper was a case report due to limited cases and study design with significant bias. I suggest the authors omit the unqualified case(such as case 1) and change the study design category during submission.
Second, as I mentioned in the previous comment, the inclusion and exclusion criteria were mandatory in the controversial issue. These criteria also help patient selection in subsequent studies. However, according to the revised manuscript, the authors enrolled patients with a known primary disease or multiple spinal lesions suggesting a metastatic process. I was not convinced by the statement of the authors in the paper regarding patient selection. We usually evaluate patients as surgical candidates by some scoring system, such as Tokuhashi score, Tomita score, ECOG performance status scale, SINS score, etc. I recommend the authors use more objective parameters or a scoring system for scientific or reproducible results.
Third, the images of cases 1 & 2 did not show post-operative results after separation surgery. Figure 9 was the T2WI image at the index level, and I did not find the enhanced T1 image to demonstrate the efficacy of "separation" after the surgery.
Fourth, there were some minor problems. For example, it is a common rule to write all non-standard abbreviations in their entirety on their first appearance in papers. For example, the acronym "MRC" lacked its entirety on its first appearance.
Lastly, I want to emphasize that the goal of cancer surgery differs from the treatment of degenerative spinal diseases. Cancer treatment is multidisciplinary and requires adjuvant therapies for disease control. Except for en-bloc resection for solitary metastasis, separation surgery can be an alternative for relieving compression of neural structure and improving functional ability. I agree that endoscope-assisted surgery is beneficial in decreasing surgical burden. However, we need to focus on the clinical and functional outcomes of patients but not only on blood loss or wound size. Also, this essential point was not mentioned in the short case report.
Author Response
1. We totally agree thet from that beginning it's case series report, we've changed the category. We don't want to omit the first case because of its didactic value- it was clinical disaster and it should be a warning for future attempts. The second case is, in turn, an example of use in a palliative patient who decides only for a minor procedure to reduce symptoms. In our opinion, this is a valuable qualifying example
2. Absolutely true, it is standard at our department to evaluate cancer patients with SINS and Bilsky score. The fact that this was not included in our description is our omission. We've corrected it as well.
3. Please note Figure 1 is T2 before the surgery and Figure 2 is T2 after the surgery. In second case because of the patient condition we did not perform MRI images that's why it's only CT. In the final case the control MRI was performed without contrast enhancement (hospital policy, short time between the examinations, elderly patient, poor kidneys), but in our opinion the result is clearly visible in T2- before the surgery the medulla is pushed to the left side, in control it is located in the center, and the CSF is visible around it.
4. Corrected
5. 100% agree on that. We have emphasized this in the conclusion.